# The Fate of the Chlorophyll Derivatives in Olives Preserved and/or Packaged in Presence of Exogenous Copper

**DOI:** 10.3390/molecules28104250

**Published:** 2023-05-22

**Authors:** Fausta Natella, Gianni Pastore, Altero Aguzzi, Paolo Gabrielli, Nicoletta Nardo, Roberto Ambra

**Affiliations:** CREA (Council for Agricultural Research and Economics), Research Centre for Food and Nutrition, 00178 Rome, Italy; fausta.natella@crea.gov.it (F.N.);

**Keywords:** table olives, chlorophylls derivatives, Cu-chlorophyllins, colour adulteration

## Abstract

Chlorophyll pigments are thought to be responsible for the highly appreciated green color of unfermented Castelvetrano-style table olives, but no studies have considered the effects of a controlled addition of copper during storage or packaging at the industrial level. For this purpose, chlorophyll derivatives were analyzed in Nocellara cultivar table olives debittered industrially using the Castelvetrano method, via means of HPLC and MS analyses, following the addition of copper in alkaline brines stored at 4 °C for 3 months in 220 L barrels, and during the subsequent storage in acid brines in commercial 400 g packages at 4 °C for up to 18 months. The presence of copper in storage or in packaging brines both contributed significantly to maintaining the green color of the olives, which was associated with a specific pattern of chlorophyll derivatives, as evidenced by principal component analysis. Notably, re-greening was rapidly achievable also for olives that had yellowed for 18 months at a copper concentration below the limit of EU legislation. Finally, by means of PCA, we also demonstrated that a short-term thermic treatment can work as an accelerated predictive tool in determining the fate of chlorophyll derivatives.

## 1. Introduction

Among the 500 species of the Oleaceae family, the olive tree (*Olea europaea* L.) is the only member that produces edible fruits and is by far the most economically important member of the family, particularly in the Mediterranean basin, which contains the majority of the world’s 1500 cultivars [1]. As regards to the production of table olives, raw fruits are not edible because of their very strong bitter taste due to the huge amount of secoiridoids, mainly oleuropein, demethyloleuropein and ligstroside, that they contain [2]. To produce table olives, various debittering methods are currently used worldwide, all aimed at hydrolysing bitter molecules to non-bitter ones, such as hydroxytyrosol, oleoside-11-methyl ester, oleoside and oleuropein aglycone [3]. Generally, these methods rely on acid or base hydrolysis of the bitter phenolic compounds and differently influence the organoleptic and nutritional properties of the edible product [4].

Because of pH changes during the debittering process, chlorophylls undergo a series of transformations that can negatively modify the color of drupes, affecting consumer choices [5]. In general, alkaline treatments determine degradation of chlorophylls a and b into more hydro-soluble derivatives [6]; dephytylation triggered by activation of the endogenous chlorophyllase; and/or isocyclic ring allomerization, yielding phytyl-chlorin or phytyl-rhodin derivatives [7]. These aspects are particularly important for the olives debittered using the Castelvetrano method, appreciated by consumers for their bright green color. In this method, olives are debittered for 2–3 weeks in an alkaline brine, without successive fermentation (typical of other debittering methods), which further negatively affects color.

Nevertheless, in order to meet consumer demand, food industries sometimes perform color adulteration of Castelvetrano-style olives via re-greening practices, either via the fraudulent addition of E141ii coloring additive (whose use is not allowed in table olives), or via equally prohibited treatment with copper salts after harvest. Finally, producers sometimes circumvent the norm that allows the agricultural practice of treating drupes before harvest with copper salts. In fact, unnecessary copper treatments of trees are frequently performed with the sole purpose to maintain the color of the drupes [7], keeping the copper content in olives just below 30 mg/kg, which is actually the maximum residue level implemented by the UE Regulation [8].

The insertion of copper ions within the chlorophyll porphyrin ring yields green metallo–chlorophyll complexes that are more stable than the original compounds due to higher metal–porphyrin bond energies, as well as being more resistant to acid and heat [9]. Actually, re-greening is a process that can also occur via the insertion of metals (Zn^2+^ or Cu^2+^) naturally present in olives; in fact, some authors have stated that an endogenous copper concentration is adequate to complex the chlorophyll derivatives to an extent sufficient enough to determine the olive re-greening [10], while others claimed that endogenous copper is not enough to achieve olive re-greening [11].

The transformations occurring to chlorophylls during Castelvetrano processing have been widely studied at the laboratory level [7,12], but no studies have considered the effects of the presence of exogenously added copper salts at the industrial scale. The aim of this work was thus to study the effects on the “colored” derivatives of chlorophylls in Castelvetrano-style olives following the addition of copper during storage or during the subsequent packaging in acidified brines at the industrial level.

## 2. Results and Discussion

### 2.1. Copper in Stored and Packaged Olives

Table 1 shows the copper content of olives debittered using the Castelvetrano method (see experimental design in Section 3.1. Materials and Methods). As expected, the endogenous copper was quite low, being 2.9 ± 0.2 mg/Kg, a value that falls within the range (1–5 mg/kg) measured for table olives [13], while it was higher in the olives stored in the presence of copper, reaching values of 23.2 ± 0.4 and 49.0 ± 0.8 mg/Kg, respectively, in barrels where 99.5 and 199 mg/L of copper were added. The addition of copper in the packaging brine further increased the endogenous copper content, and therefore obviously the highest copper concentration (80.2 ± 6.7 mg/kg) was found in the olives stored in the 199 mg of Cu^2+^/L double-enriched brine and then packaged in the brine enriched with 68 mg of Cu^2+^/L (quantification was performed 4 months after packaging, but it did not significantly change after 6 months).

### 2.2. Chlorophyll Derivatives in 3-Month-Stored Olives

Table 2 reports the content of the chlorophylls and chlorophyll derivatives in olives debittered using the Castelvetrano and stored for 3 months in barrels containing classic brine (pH 10) or in brines with increasing amounts of copper salts (99.5 or 199 mg/kg). A representative picture of the olives is also shown in the table, demonstrating that copper salts induced a marked greening. Accordingly, Cu derivatives were detectable in olives stored in Cu-enriched brines, without significant differences between the two different concentrations of Cu^2+^, while no Cu derivatives were present in drupes stored in Cu-free brine, confirming that low levels of endogenous copper are not sufficient enough to induce re-greening via complexation with chlorophylls [11]. The molecular structures of chlorophyll derivatives are shown in Figure 1.

As regards to Cu-free chlorophyll derivatives, their profiles in olives stored in the classic brine were similar to that already reported for NaOH-debittered olives [6] and were not affected significantly by copper salts (61.0 vs. 64.0 vs. 72.1 mg/Kg, Table 2), even if opposite effects were found for chlorophylls a and b as well as for pheophytinized chlorophyll derivatives. Specifically, the amounts for chlorophylls a and b were halved compared to olives stored in Cu-enriched brines, which was consistent with their fading. Notably, this halving in the olives was concomitant with an increase in the amounts of chlorophylls a and b in the brine, presumably because of a weak but significant loss of the texture of the drupes that were selectively stored in the Cu-free brine, i.e., 9.1 ± 2.6 versus 10.4 ± 2.0 and 11.9 ± 3.5 N/cm^2^, respectively, in barrels 2 and 3 (*p* < 0.005, grouped using the Tukey test). On the other hand, as shown in Table 2, a greater content of pheophytinized chlorophyll derivatives (15^2^-Me-phytyl-chlorin *e*_6_ ester, 15^2^-Me-phytyl-rhodin *g*_7_ ester, pheophytin a, pheophytin b, pyro-pheophytin a, pheophorbide a) was present in olives from the Cu-free barrel, which is somehow unexpected as pheophytinization is favored in acidic conditions (far from Castelvetrano conditions). However, a certain level of pheophytinization can still be fostered by the heat produced during the alkaline treatment of drupes [6], yielding pheophytins a and b and 15^2^-Me-phytyl Rhodin *g*_7_ ester and 15^2^-Me phytyl Chlorin *e*_6_ ester. In fact, the latter are specifically associated with alkali processing [6] and originate from Mg-15^2^-Me-phytyl Chlorin *e*_6_ ester and Mg-15^2^-Me-phytyl Rhodin *g*_7_ ester, that in turn originate from allomerization reactions on the isocyclic ring of chlorophyll a and b, respectively, thanks to the activity of endogenous chlorophyllase [14].

### 2.3. Chlorophyll Derivatives in Olives during 18 Months of Packaging

Figure 2 shows the fate of the chlorophyll derivatives in olives packaged in traditional or Cu^2+^-enriched brine and stored at 4 °C for up to 18 months. To allow a clearer view of changes, the different derivatives were grouped according to the presence of Mg (Figure 2A), 2H (Figure 2B) or Cu (Figure 2C) in their porphyrin ring (the fate of individual molecules is reported in Appendix A.

The compounds still containing Mg (chlorophyll a and b) (Figure 2A) suffered from a strong decrease in the first 4 months of conservation and then continued to slowly decline, reaching a plateau around month 12, regardless of the presence of copper in packages. This decrease is due to the transition from an alkaline brine to an acidic one that favors the substitution of Mg with H in the porphyrin ring (pheophytinization process), leading to an increase in pheophytinized molecules (Figure 2B) or copper-containing derivatives (Figure 2C), depending on the addition of copper in the packaging and storing brines. Specifically, pheophytinized molecules that were already increased following storage in the Cu-free brine, continued to slightly increase when the olives were packaged in the acidic Cu-free packaging brine (solid green lines in Figure 2B). This further increase is due to the continuation of the pheophytinization process, which is favored by the acidic environment of the packaging brine and thus explains the further yellowing of the olives observed after 18 months of storage (Figure 3). Differently, packaging in copper-enriched brines was associated with a sudden decrease in pheophytinized molecules (dotted green line in Figure 2B). This decrease was accompanied by an increase in Cu-containing derivatives (dotted green line in Figure 2C), confirming that when copper is present, it substitutes hydrogen in the porphyrin ring of the pheophytinized molecules. Notably, the accumulation of copper derivatives was constant for at least 15 months and increased during the first 2 months significantly more than after the 3 months of storage in the Cu-enriched brine barrels. The acid condition of the packaging brine in fact favors the substitution of Mg with H—a necessary step for subsequent copper insertion. On the other hand, according to the very low amount of endogenous copper in the olives (2–3 mg/kg, Table 1), no copper derivatives were ever detected in the Cu-free stored olives packaged with Cu-free brine (Figure 2C).

With respect to olives coming from Cu-enriched barrels (where formation of copper derivatives was prevented by the alkaline conditions inhibiting pheophytinization), the switch to acid packages was associated with a time-dependent increase in copper derivatives as well as in Cu-free packages, proportional to the content of copper in the drupes (Table 1 and Figure 2C, solid blue line vs. solid red line). Notably, the presence of Cu in the packages induced an equivalent increase in copper derivatives in the olives that were stored in the Cu-free barrel and in those that were stored in the double-enriched Cu barrel, possibly because most of the molecules were already pheophytinized and thus ready for copper inclusion (Figure 2). Specifically, Cu-15^2^-Me-phytyl-chlorin *e*_6_ ester, Cu-15^2^-Me-phytyl-rhodin *g*_7_ ester, Cu-pheophytin a, Cu-pyropheophytin a and Cu-pyropheophorbide a were the Cu derivatives associated with the observed re-greening phenomenon (Appendix A).

The chemical mechanism of the formation of chlorophyll derivatives during storage in barrels and in packaging is schematized in Figure 4. As shown in the figure, allomerization reactions yield phytyl-chlorin (originated from chlorophyll a) or phytyl-rhodin (originated from chlorophyll b) derivatives. In parallel, the loss of magnesium in the chlorophylls generates pheophytins that, in turn, may be dephytylated, thus yielding pheophorbides, and eventually successively decarbometoxcylated, thus yielding pyropheophorbide or directly decarbometoxcylated in pyropheophytin. In the presence of copper, all these molecules may include the metal in their porphyrin ring, yielding Cu 15^2^ Me phytyl rhodin *g*_7_ ester, Cu 15^2^ Me phytyl chlorin *e*_6_ ester, Cu pheophytin, Cu pyropheophytin and Cu pyropheophorbide.

### 2.4. Principal Component Analysis

We sought to produce a comprehensive view of the copper-induced re-greening process through a principal component analysis (PCA) performed on the mean concentrations of identified compounds grouped according to the time of conservation (fresh: 2–4 months; medium: 6–9 months; seasoned: 12–18 months) and the amount of copper at the time of the sampling (low: ≤30 mg/kg; medium: 31–60 mg/kg; high: >60 mg/Kg) (Figure 5).

The first two components (accounting for 67.29% of the total variance) indicate a clearly different pattern between the low copper (green dots) samples in quadrants one and four of the score plot (Figure 5A) and those with medium and high copper (respectively blue and red dots) quantities in quadrants two and three, with the latter (red dots) having more negative values with respect to the *x*-axis. Analogously, a clear trend appears on the *y*-axis, determined by the time of conservation, with the fresh olive samples (lighter dots) in quadrants one and two of the plot and the aged olives (darker colours) in the negative quadrants of the *y*-axis. The loading plot (Figure 5B) shows a wider view of the process and emphasizes that the components that specifically contribute to component 2 are on the one hand chlorophylls a and b (that both decrease during the 18 month conservation period), and on the other hand the time that, in the absence of copper, determines the loss of magnesium-yielding pheophytins a and b, pheophorbide a, 15^2^-Me-phytyl-chlorin *e*_6_ and 15^2^Me-phytyl-rhodin *g*_7_ esters (in the right side of the loading plot), or, when copper is present, their copper derivatives (Cu-15^2^-Me-phytyl-chlorin *e*_6_ and Cu-15^2^Me-phytyl-rhodin *g*_7_ esters, Cu-pyropheophytin a, Cu-pheophytin a and Cu-pyropheophorbide a), which are all localized in the left side of the plot, suggesting that monitoring this pattern of compounds may be predictive of copper presence and of the olive aging process.

### 2.5. Incubation of Olives at 45 °C

In an attempt to test whether it is possible to predict the formation of chlorophyll copper derivatives, olives stored for 3 months in barrels were heated at 45 °C in a stove for 10, 20 or 30 days. After 10 days, the total amount of copper derivatives was already 106–150% of that found after 18 months of conservation at 4 °C, with values that increase by 120–170% at the 30th day of incubation.

The predictive capacity of the heating test was analyzed using a PCA. The score plots of the PCA are shown in Figure 6A, where the small dots are relative to samples at the 18th month of conservation at 4 °C, whereas the bigger squares represent the samples that were just packaged and then kept on a stove for 10, 20 and 30 days.

Taking into consideration that the small green, red and blue dots (that represent the samples stored at 4 °C) are all darker as they get older (the darker samples are those older than 12 months), the trend (indicated by the arrows) clearly indicates how the low-copper samples tend to slide in time from cartesian dial 1 to dial 4 (green arrow), exactly toward the direction of the squares of the heated samples (large green squares), while the samples with medium and high copper content (red and blue small dots) tend to pass from dial 2 to dial 3, towards the squares relative to the corresponding heated samples.

### 2.6. Regreening

Besides the capacity of copper to prevent the loss of the green color throughout a preventive treatment, we also tested the possibility that copper could induce an effective re-greening of the drupes that had already lost their green color. Treatment of faded olives, already conserved for 18 months at 4 °C, through the addition of 30 and 60 mg of copper per kg, determined a proportional increase in Cu derivatives (Cu-15^2^-Me phytil Rhodin *g*_7_ ester, Cu15^2^-Me phytil chlorin *e*_6_ ester and Cu pheophytin) (Figure 7). Notably, the increase in Cu derivatives was associated with the re-greening of drupes. Similar attempts have already been performed, but using non-industrial conditions or huge amounts of copper [13,15]. Nevertheless, even if they did not mention olive color variations, Harp et al. [13] observed a similar increase in total Cu derivatives (from 0.2 to >10 mg/kg). Neither did Negro et al. mention color variations; however, they identified Cu chlorin *e*_6_ and Cu pyropheophorbide a in treated olives [15]. Our results indicate that re-greening of faded olives can already occur at 4 °C and at concentrations below the limit of EU legislation (30 mg/Kg Cu^2+^) [8]—a fact that could be considered for legal and health aspects.

## 3. Materials and Methods

### 3.1. Experimental Design

The experimental design is shown in Figure 8. Olive drupes (390 kg, Nocellara cultivar) debittered using the Castelvetrano method (for two weeks) were divided into three plastic barrels (220 L volume). The brines (80 L) of two barrels were added, respectively, with 7.97 (low Cu) and 15.94 g copper (high Cu) using CuSO_4_·5H_2_O. After 3 months, olives were split into almost one thousand commercial plastic packages each containing 400 g of olives immersed in 250 mL of 4% NaCl brine at a pH of 3.0 (acidified with tartaric acid E334 and ascorbic acid E300). In half of the packages, the brine was enriched with copper (68 mg/L of CuSO_4_·5H_2_O), thus yielding 6 different samples: storage brine/packaging brine, storage brine/Cu packaging brine; low Cu storage brine/packaging brine; low Cu-storage brine/Cu packaging brine; high Cu storage brine/packaging brine; high Cu storage brine/Cu packaging brine. In order to test the possibility of predicting the formation of copper chlorophyllins, subsamples of the 6 samples were maintained in an incubator at 45 °C for 10, 20 or 30 days. Except for T0 and heated subsamples that were processed immediately, all the samples were stored in a refrigerated room maintained at 4 °C and processed after 2, 4, 6, 9, 12, 15 and 18 months.

Finally, copper-induced re-greening was tested also after 18 months of storage at 4 °C on Cu-free olives, adding 12 and 24 mg of Cu^2+^ in each packaging (containing 400 g of olives and 250 mL of brine) and extending the incubation at 4 °C for 10, 20 or 40 days.

For all analyses, each subsample included 20 olives that were destoned, frozen in liquid nitrogen and pulverized using an IKA A 11 Basic mill (IKA, Staufen, Germany). Powders were stored at −80 °C until analyses were performed.

### 3.2. Chemicals and Standards

N,N-dimethylformamide, methanol, tert-methyl butyl ether, acetic acid, hexane, acetone, CuSO_4_, and NaCl were obtained from Carlo Erba (Milan, Italy).

Analytical standards of chlorophyll a (CAS 479-61-8) and b (CAS 519-62-0) from Merck (Darmstadt, Germany) and of pyropheophorbide a (CAS 24533-72-0), chlorin *e*_6_ (CAS 19660-77-6), rhodin *g*_7_ sodium salt (CAS 84581-15-7), pheophorbide a (CAS 15664-29-6), Cu-pheophorbide a and Cu-chlorin *e*_6_ (CAS 19660-77-6) from Livchem Logistic GmbH (Frankfurt, Germany).

### 3.3. Analyses

Dry matter was quantified gravimetrically, maintaining ≈4 g of olive pulp at 105 °C for 24 h.

Chlorophyll derivative extraction was performed according to the method of Gandul-Rojas et al. [5], although slightly modified. In brief, ≈5 g of olive pulp powder was vigorously shaken in 30 mL of N,N-dimethylformamide (orbital motion 300 rpm, 25 mm orbital) for 30 min on a Certomat MO II shaker (Sartorius, Gottingen, Germany) and then centrifugated (4000 rpm) for 3 min. The extraction was repeated twice. The extracts were defatted twice with 20 mL of n-exane, centrifugated (4000 rpm for 3 min) and joined in a separatory funnel (after being grossly filtered through a gauze) with 30 mL of diethyl ether/n-exane 1:1 and 90 mL of NaCl 10% at 4 °C. Funnels were shaken for 1 min. After settling for 30 min, the aqueous phase was discarded, and the extracts were filtered through anhydrous Na_2_SO_4_ and evaporated in a 50 mL flat-bottomed flask using a rotavapor at 35 °C, resuspended in 1.5 mL of acetone and filtered with 0.2 µm syringe filters for HPLC analyses. All the procedures were executed under green light.

Chlorophyll derivatives were detected and quantified via HPLC analyses (1200 Series, Agilent, Santa Clara, CA, USA) using a YMC-Pack C-30 (YMC Co., Ltd, Kyoto, Japan) column (4.6 ID × 250 mm; S-5 µm) and kept at 25 °C with a flow rate of 0.75 mL/min. The mobile phase consisted of (A) methanol/H_2_O/acetic acid 90:10:05 and (B) ter-methyl buthyl ether/methanol/H_2_O 100:10:0.5, running with the following gradient (64 min): 100% A then to 50% A in 43 min and then to 38% A in 1 min, maintaining 38% A during 20 min. Conditioning between runs was 14 min (modified from [15]).

Chlorophyll a and b, pyropheophorbide a, chlorin *e*_6_, rhodin *g*_7_, pheophorbide a, Cu pheophorbide a and Cu chlorin *e*_6_ were identified according to the retention times and spectra as compared with the standard bought from Merk and LiVchem Logistic GmbH (Frankfurt Germany) and quantified through calibration curves.

For the identification of other chlorophyll derivatives, HPLC peaks were individually collected and infused into an Applied Biosystems AB Sciex 3200 QTrap mass spectrometer (Foster City, CA, USA). The 3200 ESI source was tuned using the direct infusion of a standard solution of chlorophyll b at a flow rate of 10 µL/min. The optimized parameters were: declustering potential −52 eV, entrance potential −4.8 eV, collision energy −25 eV and collision cell exit potential −3 eV. Data were acquired in the negative ion MS and MS/MS modes. Identification was achieved via comparison of MS/MS spectra with those reported in the literature [13,16]. Figure 9 shows, as an example, the HPLC chromatogram at 430 nm obtained from one olive sample stored in the presence of copper.

Quantification of the molecules was performed using the corresponding calibration curve. For pigments with chlorin-, Cu-chlorin- and rhodin-type structures, calibration curves obtained for chlorine *e*_6_ and rhodin *g*_7_ sodium salt were used. For the remaining pigments, the calibration curve obtained for the parent chlorophyll was used.

Copper was quantified using inductively coupled plasma optical emission spectrometry (Optima 8000™ ICP-OES, Perkin-Elmer, Waltham, MA, USA) after liquid ashing in a microwave digestion system (1200 Mega, Milestone srl, Sorisole, Italy).

The texture of the olives was measured using a Texture Analyzer (TA.XT2, Texture Technologies Corp., New York, NY, USA). Drilling was performed using a 3-mm diameter needle bit with a head speed of 5 mm/s. The force needed to drill the olive pulp was taken as a measure of the consistency and expressed in Newton (N).

Texture has been measured for 16 drupes per sample. Copper analyses were performed in triplicate, chlorophyll derivatives analyses have been performed in duplicate.

## 4. Conclusions

In this study, individual and total chlorophyll derivatives and total elemental copper contents were measured in Castelvetrano table olives produced on an industrial level during both the barrel and packaging stages for up to 18 months. The endogenous copper of the resulting olives was quite low and did not yield a natural regreening. Accordingly, olives stored and packaged without copper underwent continuous yellowing and were void of detectable Cu derivatives, which instead rapidly increased when olives were “treated” with brines containing copper, either in barrels or in packages. Actually, the increase in Cu derivatives was faster and/or higher when copper was added in the packaging brine, thanks to the acidic conditions that favor the pheophytinization process necessary for copper insertion. Moreover, our study demonstrates that copper addition is not only able to “maintain” the brilliant green color of Castelvetrano olives during the mid-long-term conservation period, but is also effective on already yellowed ones, conserved at 4 °C for 18 months; a fact that should be considered especially for legal aspects, as copper was effective also at a concentration below the limit of EU legislation (30 mg/Kg Cu^2+^).

The copper derivatives associated with the observed re-greening phenomenon were Cu-15^2^-Me-phytyl-chlorin *e*_6_ ester, Cu-15^2^-Me-phytyl-rhodin *g*_7_ ester, Cu-pheophytin a, Cu-pyropheophytin a and Cu-pyropheophorbide a. Accordingly to what suggested by Gandul-Rojas and Gallardo-Guerrero [7], copper addition did also yield Cu-chlorophyll derivatives of the b series, whose presence could be considered a marker of deliberate copper addition because of their lower chelation reactivity for transition metals, even if an accurate indicator threshold of deliberate addition of copper is still missing. The detection of copper derivatives is further complicated by the fact that Cu derivatives are present also in the E141(i) colorant, especially Cu-pyropheophytin [17], also fraudulently used in olives as a food additive.

As indicated by a PCA, clearly different patterns for the fate of Cu-chlorophyll derivatives in olive samples with low, mid and high levels of copper were evidenced during olive aging. Finally, via a PCA, we demonstrated that the incubation of olives at 45 °C for 10 days can be a fast and highly predictive tool of for determining the fate of chlorophyll derivatives during conservation.

Despite the obvious advantage of not having mimicked industrial storage methods with laboratory scaled procedures, further studies are needed to verify the generalizability of the re-greening effect of copper on olives produced by other debittering methods.

## Figures and Tables

**Figure 1 molecules-28-04250-f001:**
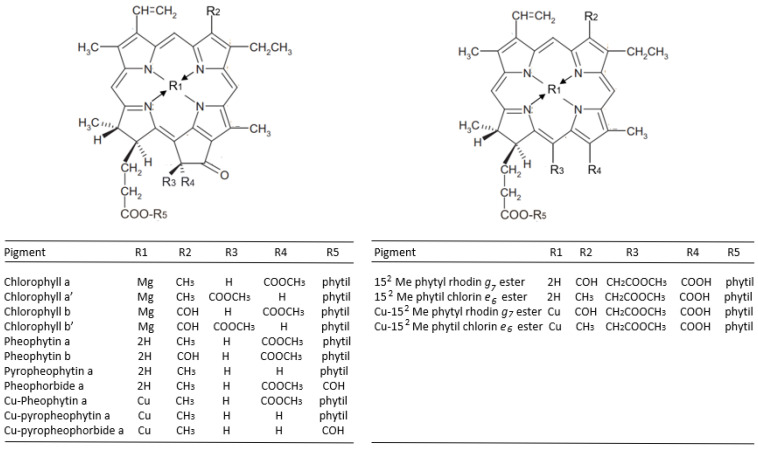
The molecular structures of the chlorophyll derivatives found in the Castelvetrano olives.

**Figure 2 molecules-28-04250-f002:**
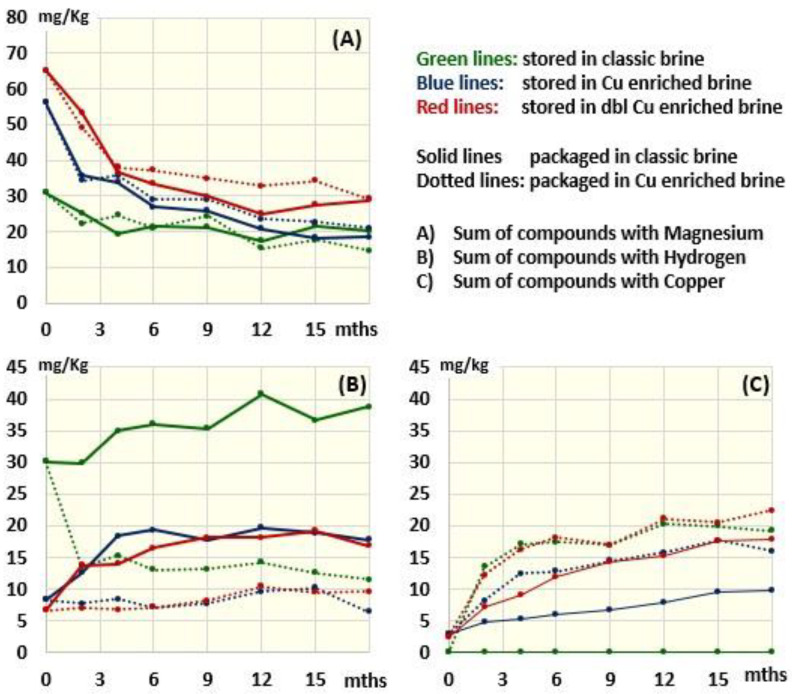
Time trend of the concentration of compounds containing Mg, H and Cu as a function of the composition of the storage and packaging brine.

**Figure 3 molecules-28-04250-f003:**
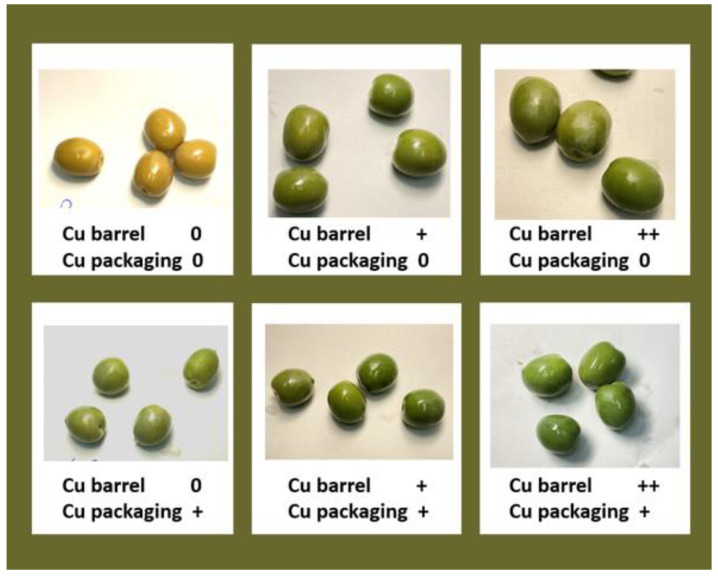
Olives stored in barrel and packages in brines without Cu (0) or in with 99.5 mg/L (+) or 199 mg/L (++) of Cu.

**Figure 4 molecules-28-04250-f004:**
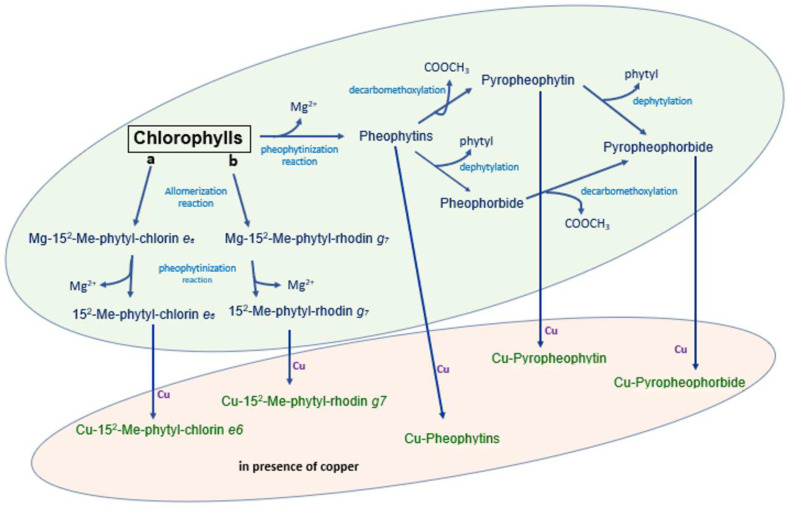
Chemical mechanism of the formation of chlorophyll derivatives during the storage of Castelvetrano olives.

**Figure 5 molecules-28-04250-f005:**
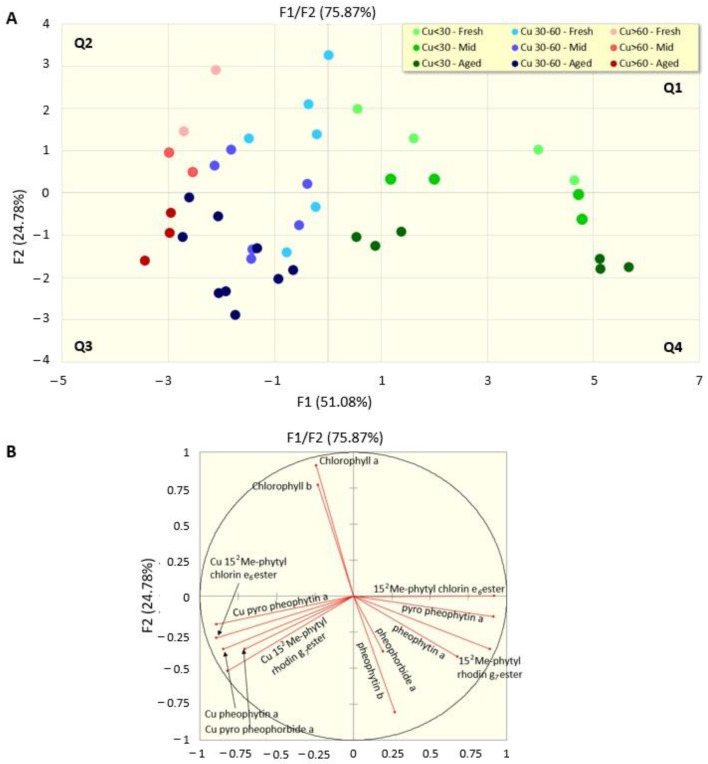
Score (**A**) and loading (**B**) plots of the principal component analysis on data grouped based on the content of copper in the drupes (<30; 30–60; >60 mg Kg) and the time of conservation in the packaging (fresh: ≤4 months; mid: 6–9 months; aged: ≥12 months).

**Figure 6 molecules-28-04250-f006:**
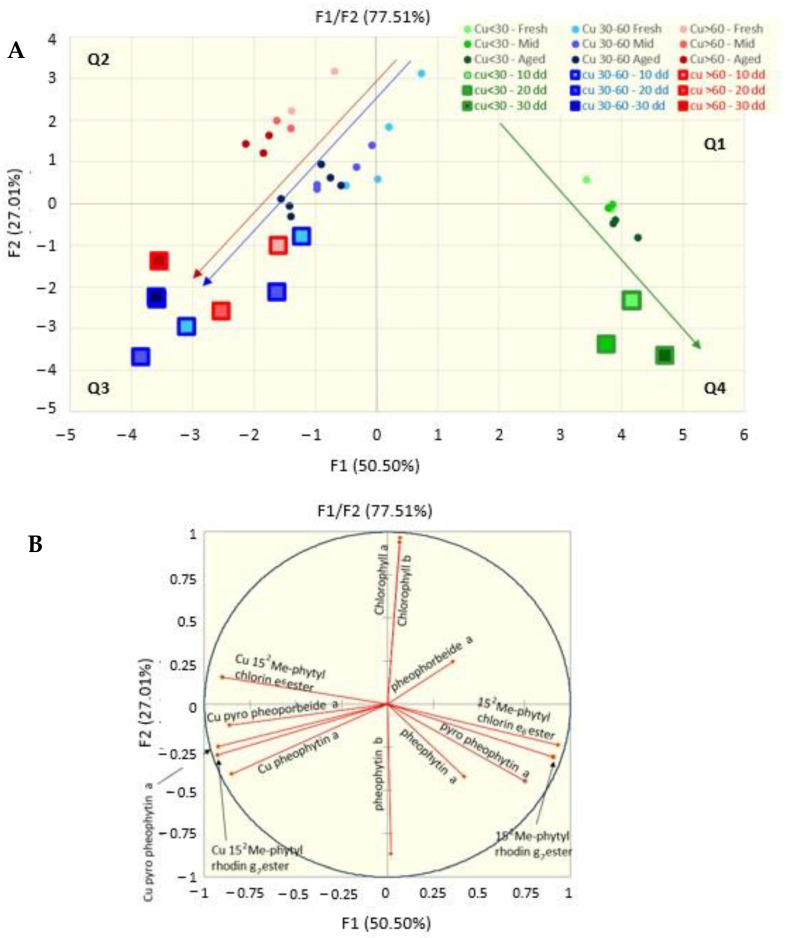
Score (**A**) and loading (**B**) plots of a principal component analysis on data grouped on the basis of the content of copper in the drupes and the time of conservation in the packaging, as compared to samples incubated at 45 °C for 10, 20 and 30 days.

**Figure 7 molecules-28-04250-f007:**
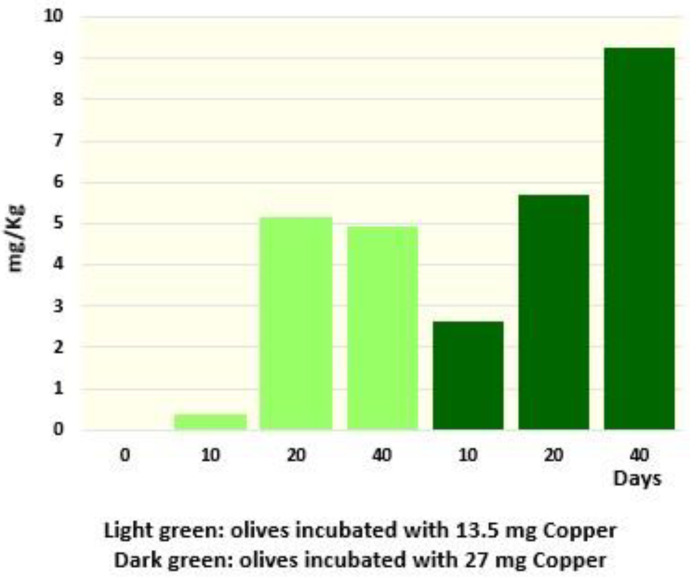
Increase in the total amount of Cu-containing molecules after 10, 20 and 40 days following the addition of 13.5 or 27 mg of copper in the packaging containing 400 g of olives and 250 mL of brine.

**Figure 8 molecules-28-04250-f008:**
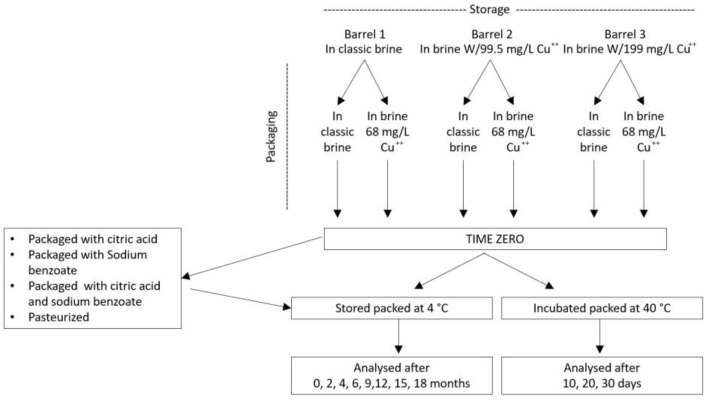
Experimental design.

**Figure 9 molecules-28-04250-f009:**
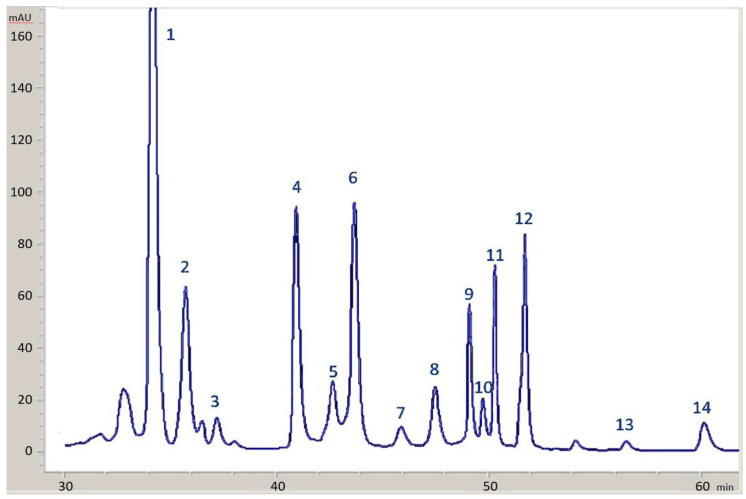
HPLC chromatogram at 430 nm for one olive sample stored in the presence of copper. Peaks: (1) Lutein; (2) Chlorophyll b; (3) Chlorophyll b’; (4) Chlorophyll a; (5) 15^2^ Me phytyl rhodin *g*_7_ ester; (6) Cu-15^2^ Me phytyl rhodin *g*_7_ ester; (7) 15^2^ Me phytyl chlorin *e*_6_ ester; (8) Cu-15^2^ Me phytyl chlorin *e*_6_ ester; (9) pheophytin b; (10) pheophorbide a; (11) pheophytin a; (12) Cu-pheophytin a; (13) pyropheophytin a; (14) Cu-pyropheophytin a.

**Table 1 molecules-28-04250-t001:** Copper content of the olives (mg/Kg) after 3 months of storage in barrels and after a further 4 months of packaging.

		After 4 Months of Packaging Packaging Brine	
	After Storage in Barrels	Classic	Enriched 68 mg Cu/L
**Storage brine**			
Classic	2.92 ± 0.23	2.17 ± 0.04	39.76 ± 1.29
Enriched 99.5 mg Cu/L	23.15 ± 0.39	21.85 ± 1.28	61.91 ± 1.64
Enriched 199 mg Cu/L	48.99 ± 0.80	38.50 ± 0.56	80.19 ± 6.74

**Table 2 molecules-28-04250-t002:** Chlorophylls and their derivatives at the end of the storage period.

	** 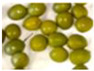 **	** 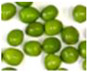 **	** 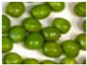 **
	**Stored in Classic Brine**	**Stored in 99.5 mg/L Cu** **Enriched Brine**	**Stored in 199 mg/L Cu** **Enriched Brine**
Chlorophyll a	14.7	26.4	30.3
Chlorophyll b	16.2	29.2	35.0
15^2^ Me phytyl rhodin *g*_7_ ester	7.8	nd	nd
15^2^ Me phytyl chlorin *e*_6_ ester	12.4	2.0	0.8
Pheophytin a	3.9	1.8	1.8
Pheophytin b	0.6	nd	nd
Pyropheophytin a	0.8	0.3	0.2
Pheophorbide a	4.6	4.3	4.0
**Total non-Cu molecules**	**61.0**	**64.0**	**72.1**
Cu 15^2^ Me phytyl rhodin *g*_7_ ester	nd	0.2	0.1
Cu 15^2^ Me phytyl chlorin *e*_6_ ester	nd	1.7	1.8
Cu pheophytin a	nd	0.5	0.4
Cu pyropheophytin a	nd	0.1	0.1
Cu pyropheophorbide a	nd	0.4	0.3
**Total Cu derivatives**	**nd**	**2.9**	**2.7**

## Data Availability

The methodology, and all the elaboration generated and used for this review, are fully available upon request from the corresponding author.

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
