# Peer review of "The Fate of the Chlorophyll Derivatives in Olives Preserved and/or Packaged in Presence of Exogenous Copper"

_molecules, 2023, doi:10.3390/molecules28104250_

Round 1
Reviewer 1 Report
The Manuscript entitle “The fate of the chlorophyll derivatives in olives preserved
and/or packaged in presence of exogenous copper”. The MS provides some interesting insights into the fate of chlorophyll derivatives in olives preserved and/or packaged in the presence of exogenous copper. Overall, the manuscript effectively summarizes the findings of the study and provides additional insights into the implications of your results. Here are a few comments:
1. Introduction: The abstract does not provide a clear and concise introduction that outlines the background of the study, the research problem, and the objectives of the research. You should consider revising the introduction to provide a more detailed context for the research.
2. Methodology: The abstract provides some details about the methodology, such as the storage conditions and the use of copper in the brine. However, it would be helpful to include more information about the experimental design, data collection, and data analysis procedures.
3. Results: The abstract presents some interesting results related to the fate of chlorophyll derivatives in olives preserved and/or packaged in the presence of exogenous copper. However, the results are not presented in a clear and concise manner. Consider revising the abstract to provide a more structured presentation of the results.
4. Significance: The abstract briefly mentions the significance of the research, such as the use of copper to maintain the color of the olives. However, it would be helpful to provide more information about the practical implications of the research.
5. The conclusion provides a clear summary of the key findings, including the effect of copper addition on the formation of Cu-derivatives and the regreening of olives.
- The mention of the PCA analysis is useful in highlighting the different patterns observed in samples with varying levels of copper and the potential predictive value of the thermic test.
- It would be helpful to include a brief discussion of the limitations of the study and potential areas for future research. For example, you could consider discussing the generalizability of your findings to olives produced using different methods or the need for additional research into the safety and regulatory implications of copper addition.
8. Language and style: The MS uses technical language and terminology that may be difficult for some readers to understand. Consider revising to make it more accessible to a broader audience.
Author Response
Please find below a point-by-point response to the reviewer’s comments.
- Introduction: The abstract does not provide a clear and concise introduction that outlines the background of the study, the research problem, and the objectives of the research. You should consider revising the introduction to provide a more detailed context for the research.
The abstract has been thoroughly revised and its introduction now contains a clearer explanation of the research and its objectives (please see the first 3 lines). The introduction section has been adapted accordingly.
- Methodology: The abstract provides some details about the methodology, such as the storage conditions and the use of copper in the brine. However, it would be helpful to include more information about the experimental design, data collection, and data analysis procedures.
Some important information on analysis procedures that were actually missing (i.e. the olive cultivar and the HPLC, MS techniques used) have been added in the abstract.
- Results: The abstract presents some interesting results related to the fate of chlorophyll derivatives in olives preserved and/or packaged in the presence of exogenous copper. However, the results are not presented in a clear and concise manner. Consider revising the abstract to provide a more structured presentation of the results.
Some too specific details have been removed from the abstract (such as the chemical name of molecules) and the results are presented in a more structured way.
- Significance: The abstract briefly mentions the significance of the research, such as the use of copper to maintain the color of the olives. However, it would be helpful to provide more information about the practical implications of the research.
More information on the practical implications of the research have been added in the abstract and throughout the text, i.e. the fact that regreening can be achieved rapidly at copper concentration below the limit of EU legislation also on yellowed olives.
- The conclusion provides a clear summary of the key findings, including the effect of copper addition on the formation of Cu-derivatives and the regreening of olives.
OK.
- The mention of the PCA analysis is useful in highlighting the different patterns observed in samples with varying levels of copper and the potential predictive value of the thermic test.
OK.
- It would be helpful to include a brief discussion of the limitations of the study and potential areas for future research. For example, you could consider discussing the generalizability of your findings to olives produced using different methods or the need for additional research into the safety and regulatory implications of copper addition.
Some aspects were added (in the Conclusions) on the need of an accurate indicator threshold of deliberate addition of copper and on findings generalizability. With respect to safety implications actually EFSA concluded that copper derivatives “safety of use as food additives cannot be assessed and the current Acceptable daily Intake (ADI) should be withdrawn” (10.2903/j.efsa.2015.4151). In any case, the content of derivatives per single “regreened” olive was less than 60 µg, very far from the ADI (15 mg kg−1 body weight/day).
- Language and style: The MS uses technical language and terminology that may be difficult for some readers to understand. Consider revising to make it more accessible to a broader audience.
Language was simplified as far as possible for such a technical subject, throughout the manuscript.
Reviewer 2 Report
The paper entitled ‘The fate of the chlorophyll derivatives in olives preserved and/or packaged in presence of exogenous copper’ reported the presence of copper and olives’ aging process. This work in interesting. I have no specific concern but it is mandatory to correct the manuscript in this point:
1. Please add the structural formula of the target chlorophyll derivatives with their CAS number.
2. Metallic elements with different valence, e.g. Zn2+, should be modified as ‘Zn2+’. Please the full manuscript.
3. Aging process of olive with copper presence is interesting, please add a possible chemical mechanism, preferably with a relevant reaction mechanism.
Author Response
Please find below a point-by-point response to the reviewer’s comments.
The paper entitled ‘The fate of the chlorophyll derivatives in olives preserved and/or packaged in presence of exogenous copper’ reported the presence of copper and olives’ aging process. This work in interesting. I have no specific concern but it is mandatory to correct the manuscript in this point:
- Please add the structural formula of the target chlorophyll derivatives with their CAS number.
Structural formulas were added (Figure 1) and available CAS numbers were added in the Materials and Methods section.
- Metallic elements with different valence, e.g. Zn2+, should be modified as ‘Zn2+’. Please the full manuscript.
Done throughout the manuscript.
- Aging process of olive with copper presence is interesting, please add a possible chemical mechanism, preferably with a relevant reaction mechanism.
Done, please see Figure 4.
Reviewer 3 Report
This manuscript deals with the addition of copper salt in some of the processing steps of Castelvestrano style table olives. This subject has been the research line of Gandul-Rojas and colleagues so lacks novelty and originality. Authors should consider to mention in the introduction whether addition of Cu++ in the processing of table olives is allowed or not. This point is quite significant to evaluate the significance of the content and enhance quality of presentation.
Identification of chlorophyllic pigments is also a major concern regarding revision of this manuscript. Authors state conditions for identification however, no spectra are shown for structural characterization of the pigments they describe. Preparation of standards for identification have been detailed elsewhere (Minguez-Mosquera, Gandul-Rojas, Gallardo-Guerrero and Jarén-Galán "Chlorophylls". In Methods of Analysis for Functional Foods and Nutraceuticals; Hurst, W. J., Ed.; CRC Press: Boca Raton, FL, 2002; pp 159-218). Also, no HPLC chromatograms are shown to evaluate the pigment profiles before and after Cu++ treatment.
Author Response
Please find below a point-by-point response to the reviewer’s comments.
This manuscript deals with the addition of copper salt in some of the processing steps of Castelvestrano style table olives. This subject has been the research line of Gandul-Rojas and colleagues so lacks novelty and originality. Authors should consider to mention in the introduction whether addition of Cu++ in the processing of table olives is allowed or not. This point is quite significant to evaluate the significance of the content and enhance quality of presentation.
The point about the legal permissibility of copper addition raised by the referee has been added in the introduction (please see the third paragraph).
Identification of chlorophyllic pigments is also a major concern regarding revision of this manuscript. Authors state conditions for identification however, no spectra are shown for structural characterization of the pigments they describe. Preparation of standards for identification have been detailed elsewhere (Minguez-Mosquera, Gandul-Rojas, Gallardo-Guerrero and Jarén-Galán "Chlorophylls". In Methods of Analysis for Functional Foods and Nutraceuticals; Hurst, W. J., Ed.; CRC Press: Boca Raton, FL, 2002; pp 159-218). Also, no HPLC chromatograms are shown to evaluate the pigment profiles before and after Cu++ treatment.
The reference of chlorophyllic pigments spectra has been mentioned in the text (ref. 16) and one HPLC chromatogram has been added in the Materials and Methods section (Fig. 9).